# Decomposition of Naphthalene by Dielectric Barrier Discharge in Conjunction with a Catalyst at Atmospheric Pressure

**Jinjin Li [1], Zhi Zheng [1], Xiao Cui [1], Yunhu Liu [1], Ting Fan [1], Yueyue Liu [1], Dalei Chang [1,***
**and Dezheng Yang [1,2,3,*]**

[1] College of Sciences, Shihezi University, Shihezi 832003, China; lijinjin@stu.shzu.edu.cn (J.L.); zhengzhi@stu.shzu.edu.cn (Z.Z.); xiaocui78@foxmail.com (X.C.); liuyh@shzu.edu.cn (Y.L.); ft_tea@shzu.edu.cn (T.F.); yueyue_12021@163.com (Y.L.)

[2] Key Laboratory of Materials Modification by Laser, Ion, and Electron Beams, Dalian University of Technology, Ministry of Education, Dalian 116024, China

[3] Key Laboratory of Environmental Monitoring and Pollutant Control of Xinjiang Bingtuan, Shihezi University, Shihezi 832003, China

\* Correspondence: chang-da-lei@163.com (D.C.); yangdz@dlut.edu.cn (D.Y.)

**Abstract:** In this study, coaxial dielectric barrier discharge (DBD) plasma, in conjunction with a metal oxide catalyst, was used to degrade naphthalene. The characteristics of plasma discharge were studied by measuring voltage and current waveforms and the Lissajous figure. The effects of different parameters of the process on naphthalene decomposition in air were investigated. XRD, BET, and SEM data were used to investigate the nature, specific surface area, and surface morphology of the catalyst. The results show that the mineralization of naphthalene reached 82.2% when the initial naphthalene concentration was 21 ppm and the total gas flow rate was 1 L/min in the DBD reactor filled with $Al_2O_3$. The mineralization of naphthalene first increased and then became stable with the increase in treatment time and discharge power. The $TiO_2$ catalyst has more apparent advantages than the two other studied catalysts in terms of the removal efficiency and mineralization of naphthalene due to this catalyst's large specific surface area, porous structure, and photocatalytic properties. In addition, the introduction of a small amount of water vapor can promote the mineralization and $CO_2$ selectivity of naphthalene. With further increases in the water vapor, $Fe_2O_3$ has a negative effect on the naphthalene oxidation due to its small pore size. The $TiO_2$ catalyst can overcome the adverse effects of water molecule attachment due to its photocatalytic properties.

**Keywords:** DBD; metal oxide catalyst; naphthalene decomposition; mineralization

## 1. Introduction

As a result of the development of society and industry, the content of polycyclic aromatic hydrocarbons (PAHs) in the environment has increased significantly. PAHs are a form of persistent organic pollutant having different structures and toxicity levels [1,2]. Due to their carcinogenicity and persistence, PAHs in air, water, and soil have become a serious threat to human health [3]. Therefore, PAH emission has received special attention from the World Health Organization [4], and the degradation and removal of PAHs have also become the focus of scientific research [5].

As the simplest PAH, naphthalene has attracted much attention due to its strong volatility and ubiquity in the atmosphere [6]. Many methods have been applied to degrade naphthalene in the environment, such as biodegradation, catalytic oxidation, and physical adsorption [7,8]. However, the above methods have limitations in terms of decomposition time, energy efficiency, adsorption capacity, etc. [9,10]. Compared with the above methods, non-thermal plasma (NTP) technology has the advantages of high removal efficiency, short treatment time, and wide applicability [11,12]. Thus, NTP has become an essential technology for degrading naphthalene and has received widespread attention.



In NTP, the particle temperature ranges from room temperature to approximately $3 \times 10^4$ K [13]. At room temperature, high-energy electrons generated by NTP are like "electronic scissors". Through the collisions of electrons with background gases, the carrier gas molecules are dissociated, excited, and ionized to produce active particles such as excited particles, ions, free radicals, and electrons [14,15]. Then, these oxidize, reduce, or decompose pollutants, resulting in the effective degradation of naphthalene [16]. NTP can be produced using various approaches, including electron beams, corona discharge, gliding arc discharge, dielectric barrier discharge, and microwave plasma [17,18]. Dielectric barrier discharge (DBD) is one of the most commonly used methods to generate NTP in atmospheric air [19]. Wu et al. [6] reported the effect of different background gases on naphthalene decomposition in DBD. It was found that the $O_2$ concentration was a key influencing factor in the naphthalene decomposition and $CO_x$ formation. Rdeolfi et al. [20] studied the decomposition of naphthalene using a DBD reactor and found that the introduction of water vapor in the discharge could further promote the naphthalene degradation by OH atoms, which have better oxidation capacity than O atoms.

Compared with the process using plasma alone, the combination of DBD plasma and a catalyst further promotes the decomposition of gas-phase pollutants. Furthermore, it also addresses the problem of insufficient mineralization and reduces the generation of undesirable by-products. Nguyen et al. [21] used a Ag/$\alpha$-$Al_2O_3$ catalyst coupled with DBD plasma to improve the conversion of soot simulant (naphthalene). The results showed that the combined use of the catalyst and plasma solved the problem of the poor removal of naphthalene at low operating temperatures. The plasma–catalyst completely decomposed the naphthalene at 350 °C with specific input energy of 90 J/L. Wu et al. [4] used a DBD reactor driven by a nanosecond pulse to oxidize naphthalene by combining plasma and a $TiO_2$/diatomite catalyst. The study showed that the addition of the $TiO_2$/diatomite catalyst effectively improved the conversion rate of naphthalene (up to 92%) and $CO_x$ selectivity (up to 40%), and significantly reduced the formation of aerosols and secondary volatile organic compounds compared with the plasma-alone process. Therefore, it is necessary to investigate the treatment of naphthalene by combining plasma and catalysts.

The transition metal oxides, which include $Fe_2O_3$, $ZnO_2$, $TiO_2$, $CuO$, $Al_2O_3$, and MgO, can be widely used as catalysts due to their low cost, anti-toxicity, and high catalytic activity [22–24]. Choi et al. [25] mentioned that $Al_2O_3$ was usually used as a catalyst because of its thermally stable and relatively high thermal conductivity. Wu et al. [26] investigated the effect of $Fe_2O_3$ on the chemical looping generation of $NH_3$, and found that $Fe_2O_3$ can reduce the activation energy of the N-desorption step due to its considerable catalytic effect on the reaction. Bagheri et al. [27] found that $TiO_2$, a photocatalyst, is widely applied in industry due to its high physical and chemical stability, in addition to its high activity.

In this study, a plasma catalytic degradation platform was independently constructed. A uniform and stable coaxial DBD plasma, in conjunction with three transition metal oxides ($Al_2O_3$, $Fe_2O_3$, and $TiO_2$), was used to degrade naphthalene. The synergy of each catalyst with DBD was investigated based on discharge characteristics, removal efficiency, mineralization, and $CO_2$ selectivity. In this paper, the effects of initial naphthalene concentration, total gas flow rate, discharge power, treatment time, and relative humidity (RH) on mineralization and $CO_2$ selectivity in the DBD reactor filled with the catalyst are discussed. Finally, a discussion of the physical structure of the catalysts is presented, in which the specific surface area, surface morphology, and microstructure are taken into account.

## 2. Materials and Methods

### 2.1. Materials

In this study, three transition metal oxides ($Al_2O_3$, $Fe_2O_3$, and $TiO_2$) were used as catalysts to fill the discharge gap. All of these filling materials were purchased from advanced material professional manufacture Co., Ltd. Their particle sizes were 1–3 mm, and they had a purity of 99.9%. Naphthalene at a purity of 99.7% was provided by

Shanghai Macklin Biochemical Co., Ltd. The materials are commercially available and were untreated.

### 2.2. Experimental Setup

A schematic of the experimental setup used in the present study is shown in Figure 1. The gas supplied by the system was composed of synthetic air ($N_2$:$O_2$ = 4:1) (99.99% purity). The total gas flow rate was maintained at 500 mL/min and controlled by the mass flow controller (MFC). In order to ensure the stability of the gas-phase naphthalene and the water vapor input to the reactor, $N_2$ in synthetic air was separated into three paths, two of which were dedicated to carrying naphthalene and water vapor. The naphthalene vapor was generated by placing a sealed quartz tube filled with 3 g naphthalene into a 30 °C water bath pot. $N_2$ was passed through the quartz tube as the carrier gas of the naphthalene vapor, and the flow rate was set to 200 mL/min. A heating tape system (55~60 °C) was equipped in the inlet gas line to prevent the condensation of the naphthalene vapor. The naphthalene in the $N_2$ carrier gas was collected by another quartz tube filled with 5 mL of anhydrous ethanol, and the naphthalene concentration was measured by an ultraviolet spectrophotometer (UV-Vis-NIR, Agilent Technologies). A portion of the carrier gas was passed through a bubble bottle, which was placed in an insulated drum with cold water and combined with another portion of the dry gas to control the proportion of humidity.

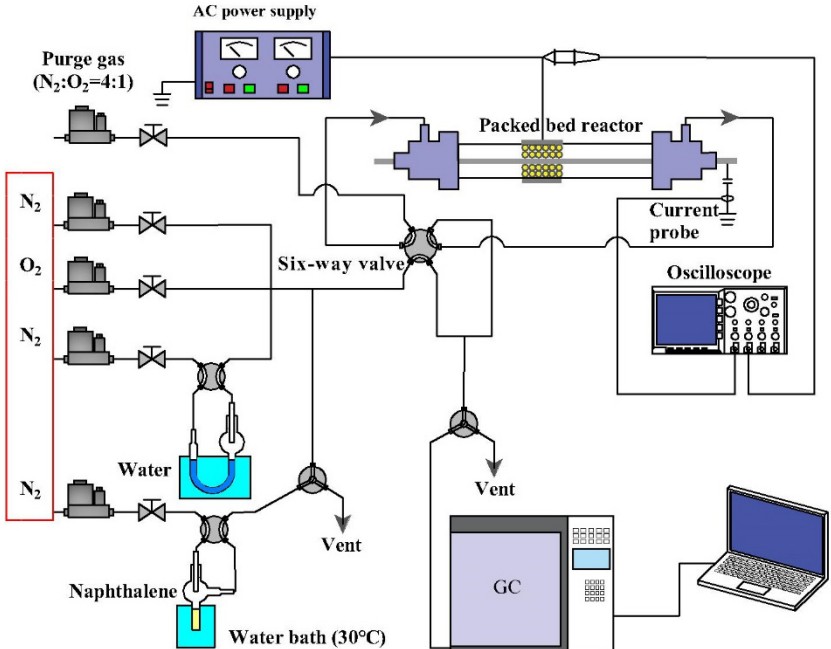

**Figure 1.** Schematic diagram of the experimental setup.

The reactor used in this work was a coaxial DBD reactor consisting of a quartz tube having an inner diameter of 12 mm, a thickness of 1.5 mm, and a length of 250 mm. A piece of silver-plated copper (25 mm height) was wrapped around the quartz tube as a high-voltage electrode. A stainless steel rod having a 6 mm diameter was inserted into the quartz tube as a low-voltage electrode. The transition metal oxides (3 g) were placed in the discharge gap with a packing length of 25 mm to constitute a plasma–catalyst reactor.

### 2.3. Testing Methods

An AC power supply (CTP-2000K, Nanjing Suman) was used to supply the DBD reactor voltage with a fixed frequency of 12.6 kHz. The applied voltage and current waveforms were measured using a 1:1000 HV probe (P6015A, Tektronix) and a current probe (TPP0500B, Tektronix), respectively. The electrical signals were recorded by a digital oscilloscope (OSC, MDO3034, Tektronix). The discharge power was determined by the product of

the discharge energy and the driving frequency. The gas from the outlet of the DBD reactor was analyzed using a gas chromatograph (GC 7900, Techcomp) equipped with a carbon molecular sieve column and a PEG-20000 capillary column, two flame ionization detectors (FID) for each column, and a methanation converter, where the methanation converter was used to measure CO and $CO_2$ concentrations. The specific surface area, microstructure, and surface morphology of the catalysts were characterized using the Brunauer–Emmett–Teller method, X-ray diffraction, and scanning electron microscopy, respectively.

The discharge power *P* (W) of the DBD reactor was calculated by Equation (1):

$$P(\text{W}) = f \times E = f \times C \times S,\tag{1}$$

where *f* is the driving frequency, *C* is the additional capacitance (0.22 μF in this study), and *S* is the integral area of the closed Lissajous figure.

The removal efficiency of naphthalene ($\eta_{\text{naphthalene}}$), mineralization ($M_{CO_x}$), $CO_2$ selectivity ($S_{CO_2}$), and $CO_x$ concentration ($C_{CO_x}$) are defined in Equations (2)–(5):

$$\eta_{\text{naphthalene}}(\%) = \left( \frac{C_{\text{in}} - C_{\text{out}}}{C_{\text{in}}} \right) \times 100,\tag{2}$$

$$M_{CO_x}(\%) = \frac{C_{CO_2} + C_{CO}}{10 C_{\text{nap}}} \times 100,\tag{3}$$

$$S_{CO_2}(\%) = \frac{C_{CO_2}}{C_{CO_2} + C_{CO}} \times 100,\tag{4}$$

$$C_{CO_x}(\text{ppm}) = C_{CO_2} + C_{CO},\tag{5}$$

where $C_{\text{in}}$ and $C_{\text{out}}$ are the concentrations of naphthalene at the inlet and outlet, respectively. $C_{CO}$ and $C_{CO_2}$ are the concentrations of produced CO and $CO_2$, respectively. $C_{\text{nap}}$ is the naphthalene concentration passing through the reactor.

## 3. Results and Discussion

### 3.1. Discharge Characteristics

Figure 2 plots the voltage and current waveforms of the DBD reactor without a catalyst and filled with an $Al_2O_3$ catalyst. The driving frequency was fixed at 12.6 kHz. As shown in Figure 2, the discharge current filaments without a catalyst at 14 kV are almost invisible because the gas was not broken down in the plasma discharge region (Figure 2a). In comparison, the gas was broken down in the discharge gap, and the discharge current filaments can be clearly observed in the DBD reactor filled with $Al_2O_3$ at 14 kV (Figure 2c). This phenomenon shows that the addition of catalysts can decrease the breakdown voltage in the discharge gap. Furthermore, the discharge current filaments and uniformity apparently increased with the introduction of the catalyst at 16 kV (Figure 2b,d). These results occurred because $Al_2O_3$ has a higher dielectric constant than gas, and the introduction of the catalyst reduces the discharge gap of gas [28]. As such, an intense electric field is created in the areas of the contact points among the catalyst pellets, which decreases the breakdown voltage of the discharge gap and increases the number of microdischarges [29]. As previously mentioned, the filling materials play a vital role in DBD. It is necessary to explore the influence of filling materials in plasma technology experiments.

Figure 2c,d shows that the number of micro-discharges and the amplitude of the current pulse increased with the increase in the applied voltage. The maximum amplitude of the current pulse increased from 5 to 7 mA. The main discharge patterns in the DBD reactor filled with $Al_2O_3$ are those of current filament discharge and surface discharge [30]. Many current pulses are generated per half-cycle of the applied voltage in the same periodicity, and the number of current pulses in the positive cycle is nearly equal to that in the negative cycle [31,32].

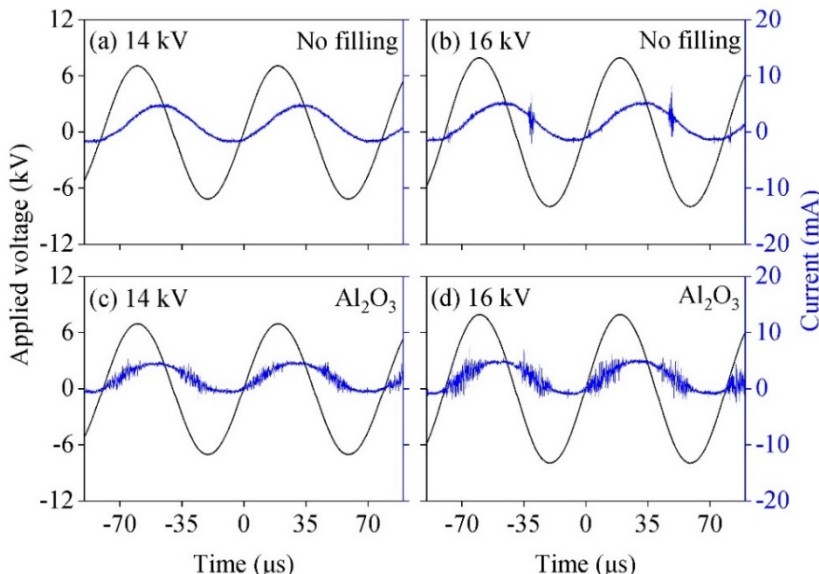

**Figure 2.** The voltage and current waveforms of discharge in the DBD reactor: (**a**) without a catalyst at 14 kV; (**b**) without a catalyst at 16 kV; (**c**) filled with $Al_2O_3$ at 14 kV; (**d**) filled with $Al_2O_3$ at 16 kV. (Driving frequency: 12.6 kHz; total gas flow rate: 500 mL/min; discharge gap: 3 mm).

An additional capacitor providing much greater capacitance that that of the DBD reactor was used in series with the discharge circuit to accumulate the total charge during one AC half-cycle; its voltage can be plotted versus the discharge voltage in the Lissajous figure [31,33]. As shown in Figure 3, the closed curve area corresponds to the energy $E_k$ consumed per cycle of the operating voltage [33,34]. Moreover, at the same input voltage, the Lissajous figure of the packing $TiO_2$ has a larger area than the other catalysts, which means that $TiO_2$ can further enhance the discharge power. From the slopes of the graph, it can be concluded that the $TiO_2$ catalyst has greater effective capacitance $C_{eff}$ [35]; that is, the $TiO_2$ catalyst has a stronger charge-transfer ability than the other two metallic oxides.

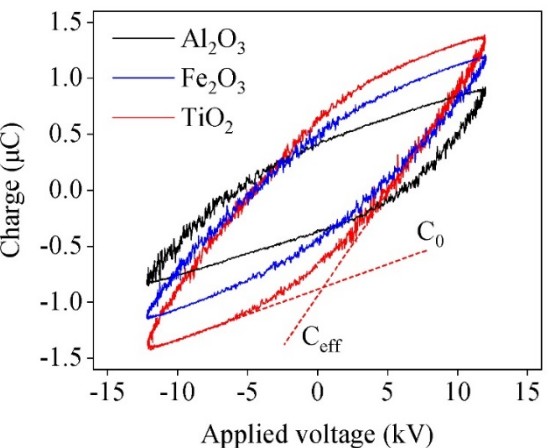

**Figure 3.** The Lissajous figures of the reactor filled with $Al_2O_3$, $Fe_2O_3$, and $TiO_2$. (Driving frequency: 12.6 kHz; peak applied voltage: 24 kV; total gas flow rate: 500 mL/min; discharge gap: 3 mm).

*3.2. The Influence of Different Filling Materials on Naphthalene Decomposition*

The removal efficiency and the mineralization of naphthalene under different filling materials are presented in Figure 4. It can be seen that the removal efficiency and the mineralization of naphthalene in the DBD reactor filled with $Al_2O_3$, $Fe_2O_3$, and $TiO_2$ are higher than those without filling materials. When the applied voltage was 24 kV, the removal efficiency and the mineralization of naphthalene in the DBD reactor without filling

materials were 76.1% and 53.4%, respectively. At an applied voltage of 24 kV, the removal efficiency of naphthalene in the DBD reactor filled with $Al_2O_3$, $Fe_2O_3$, and $TiO_2$ was 82.6%, 84.5%, and 86.8%, respectively, and the mineralization of naphthalene in the DBD reactor filled with $Al_2O_3$, $Fe_2O_3$, and $TiO_2$ was 71.8%, 66.3%, and 71.7% at 24 kV, respectively. This is because adding catalysts can increase the electric field intensity in the discharge gap via the discharge characteristics discussed in the former section [29], and promote the formation of more reactive oxygenated radicals [36]. In addition, the catalysts affect the removal efficiency and mineralization of naphthalene due to the different catalytic activity [36,37].

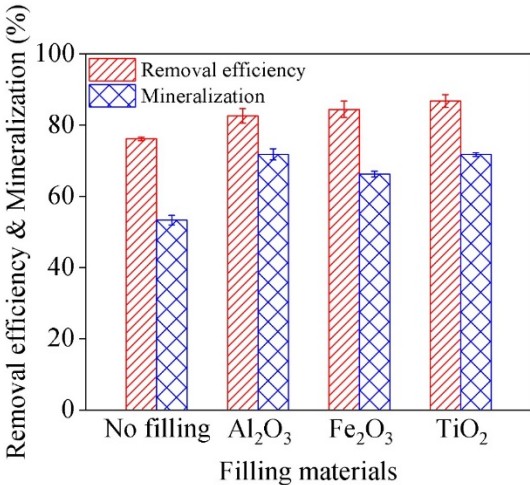

**Figure 4.** The removal efficiency and the mineralization of naphthalene under different filling materials. (Applied voltage: 24 kV; driving frequency: 12.6 kHz; initial naphthalene concentration: 21.5 ppm; total gas flow rate: 500 mL/min; discharge gap: 3 mm).

It is worth noting that carbon oxides and other by-products are produced during the degradation of naphthalene by plasma. It is difficult to analyze these by-products due to their complex composition [4,38]. The mineralization of pollutants reflects the proportion of by-products formed. The degree of mineralization is higher, which means fewer by-products are produced [39]. It is necessary to study the mineralization in pollutant decomposition.

### 3.3. The Influence of Initial Naphthalene Concentration on Mineralization

Figure 5 shows the effect of initial naphthalene concentration ($C_i$) on the mineralization and $CO_x$ concentration in the DBD reactor filled with $Al_2O_3$ at 55 W. The mineralization increased from 35.8% to 68.1% and the $CO_x$ concentration increased from 60 to 286 ppm when $C_i$ rose from 17 to 42 ppm. It is worth mentioning that, although the concentration of $CO_x$ produced remained stable, the mineralization decreased with increasing $C_i$ at higher $C_i$ ($C_i > 42$ ppm). The mineralization decreased from 68.1% to 32.8% when $C_i$ increased from 42 to 92 ppm. These results indicate that the increase in the number of naphthalene molecules per unit time is conducive to the oxidation of naphthalene within a suitable $C_i$ range [40], which increases mineralization and $CO_x$ concentration, while the naphthalene decomposition reaction competes with the oxidation of intermediate products. The oxidation of intermediate products is inhibited at higher $C_i$; thus, the $CO_x$ concentration no longer increases and the mineralization decreases as $C_i$ increases further [41]. As previously mentioned, the concentration of naphthalene plays a vital role in carbon oxide formation and the deep oxidation of naphthalene. The oxidation reaction of naphthalene ($C_{10}H_8$) is as follows [4]:

$$C_{10}H_8 + A^* \rightarrow \text{Intermediate products} \tag{R1}$$

$$\text{Intermediate products} + A^* \rightarrow CO + CO_2 \tag{R2}$$

where $A^*$ is an active species, which can be an O atom or/and •OH.

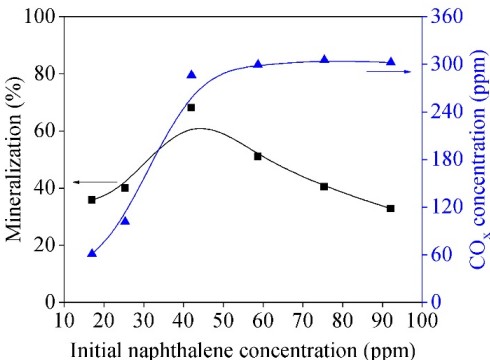

**Figure 5.** Effect of initial naphthalene concentration on the mineralization and $CO_x$ concentration in a DBD reactor filled with $Al_2O_3$. (Discharge power: 55 W; driving frequency: 12.6 kHz; initial naphthalene concentration: 21.5 ppm; total gas flow rate: 500 mL/min; discharge gap: 3 mm).

### 3.4. The Influence of Flow Rate on Mineralization and $CO_2$ Selectivity

The influence of the total gas flow rate on the mineralization and $CO_2$ selectivity in a DBD plasma reactor having a discharge power of 55 W is depicted in Figure 6. The flow rate of the naphthalene carrying gas was fixed at 200 mL/min, and the total gas flow rate of the reactor was changed by adjusting the flow rate of the other gas paths. Figure 6 illustrates that the total gas flow rate significantly affects the formation of carbon oxides during naphthalene decomposition, as evidenced by increasing mineralization from 51.6% to 82.2% when the total gas flow rate increased from 0.2 to 1.0 L/min [35]. However, the mineralization no longer significantly changed when the feeding gas was further increased. These results indicate that the total gas flow rate of 1.0 L/min is an optimum condition for the generation of carbon oxides in the DBD system. The mineralization of naphthalene reached 82.2% when the naphthalene carrying gas flow rate was 200 mL/min ($C_i$ = 21 ppm) and the total gas flow rate was 1 L/min in the DBD reactor filled with $Al_2O_3$. The gas penetration increased as the total gas flow rate increased; as a result, the gas has greater contact with the surface of the filling materials and the oxidation of naphthalene is greater [40]. This was also confirmed by the rise in $CO_2$ selectivity as the total gas flow rate increased, as shown in Figure 6. However, the gas was fully in contact with the filling materials when the gas flow rate increased to 1 L/min, and the contact area between the gas and the filling materials did not change with the further increase in the total gas flow rate. Thus, the oxidation of naphthalene was no longer significantly changed, and the mineralization tended to be relatively stable.

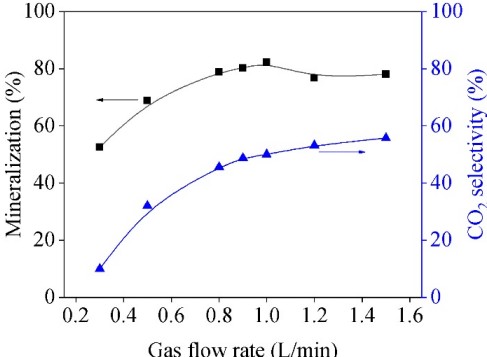

**Figure 6.** Effect of the total gas flow rate on the mineralization and $CO_2$ selectivity in a DBD reactor filled with $Al_2O_3$. (Discharge power: 55 W; driving frequency: 12.6 kHz; naphthalene carrying gas: 200 mL/min; discharge gap: 3 mm).

*3.5. The Influence of Discharge Time on Mineralization and $CO_2$ Selectivity*

Figure 7 shows the mineralization and $CO_2$ selectivity under different filling materials in the DBD reactor having a discharge power of 55 W. The mineralization and $CO_2$ first increases, and then stabilized with time. This is because some carbonaceous by-products were attached to the catalyst and enriched in the process of naphthalene decomposition; this increases the probability of collisions with the carbonaceous by-products and the oxidation with active species [42]. Furthermore, as the discharge time increases, the high-energy particles produced by plasma continuously bombard the catalyst, enhancing the desorption of the carbonaceous by-products on the catalyst. The reaction eventually achieves dynamic equilibrium through the continuous cycle of adsorption and desorption on the catalyst surface [43].

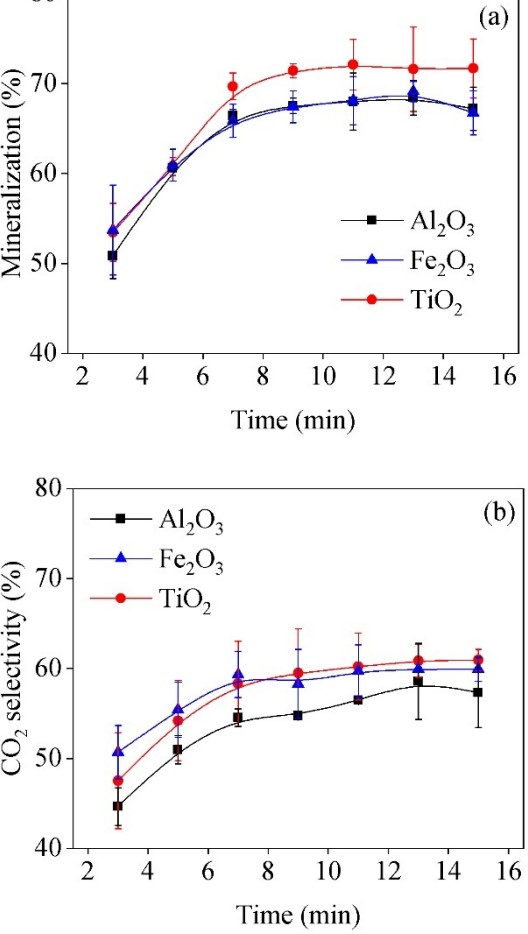

**Figure 7.** Effect of time on (**a**) the mineralization and (**b**) the $CO_2$ selectivity in a DBD reactor filled with $Al_2O_3$, $Fe_2O_3$, and $TiO_2$. (Discharge power: 55 W; driving frequency: 12.6 kHz; initial naphthalene concentration: 21.5 ppm; total gas flow rate: 500 mL/min; discharge gap: 3 mm).

As shown in Figure 7a, the mineralization in the DBD reactor filled with $Fe_2O_3$ was almost the same as that filled with $Al_2O_3$, and both were lower than that of the DBD reactor filled with $TiO_2$. However, the $CO_2$ selectivity of the DBD reactor filled with $Fe_2O_3$ and $TiO_2$ was better than that of the DBD reactor filled with $Al_2O_3$ (Figure 7b). These results indicate that $Fe_2O_3$ and $TiO_2$ have a better oxidation performance during the reaction than $Al_2O_3$. In addition, $TiO_2$ shows more apparent advantages in carbon oxide formation than the other catalysts. This may be due to the fact that $TiO_2$ is a photocatalyst, and can therefore use the ultraviolet light generated by the discharge to generate electron–hole pairs that promote the production of various strong active species, such as $\bullet OH$ and $\bullet HO_2$ [44]. Thus, the decomposition of naphthalene can be promoted. The active species can be produced

via the following reactions (see (R3)–(R9)) [44,45]. These findings indicate that the catalytic performance of filling materials plays a vital role in carbon oxide formation.

$$TiO_2 + hv \rightarrow e^- + h^+ \tag{R3}$$

$$h^+ + H_2O \rightarrow OH^- + H^+ \tag{R4}$$

$$h^+ + OH^- \rightarrow \bullet OH \tag{R5}$$

$$e^- + O_2 \rightarrow O_2^- \tag{R6}$$

$$O_2^- + H^+ \rightarrow \bullet HO_2 \tag{R7}$$

$$HO_2 + H^+ + e^- \rightarrow \bullet H_2O_2 \tag{R8}$$

$$H_2O_2 + e^- \rightarrow \bullet OH + OH^- \tag{R9}$$

### 3.6. The Influence of Discharge Power on Mineralization and $CO_2$ Selectivity

The formation of the active species is closely related to the discharge power, which is an important electrical parameter [46]. The mineralization and $CO_2$ selectivity under three different catalysts at the treatment time of 7 min are displayed in Figure 8. It can be seen that the mineralization increased with increasing discharge power when the discharge power was not higher than 30 W under the three different catalysts. The electron energy increases with an increase in the discharge power, which increases the number of active species [40] and promotes collisions between active particles and background gas molecules in the discharge zone [38]. Thus, the oxidation of naphthalene is enhanced. Furthermore, the mineralization in the DBD reactor filled with $Fe_2O_3$ was the highest. As a *p*-type multivalent metal oxide, the $Fe_2O_3$ catalyst is expected to have high activity due to its highly mobile chemisorbed oxygen [47].

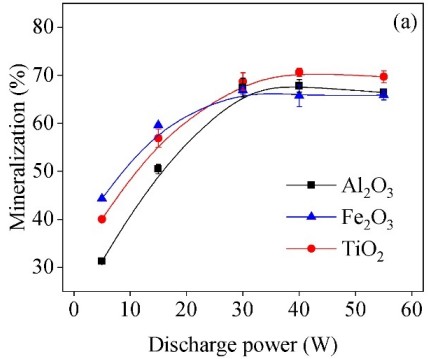

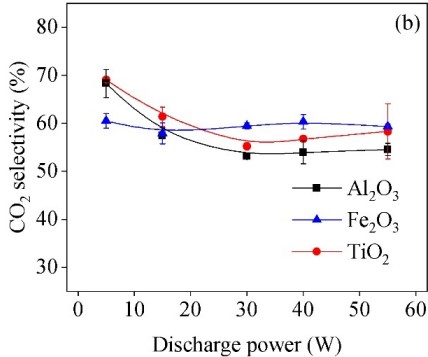

**Figure 8.** Effect of discharge power on (**a**) the mineralization and (**b**) the $CO_2$ selectivity in the DBD reactor filled with $Al_2O_3$, $Fe_2O_3$, and $TiO_2$. (Driving frequency: 12.6 kHz; initial naphthalene concentration: 21.5 ppm; total gas flow rate: 500 mL/min; discharge gap: 3 mm).

However, when the discharge power was not higher than 30 W, the $CO_2$ selectivity decreased with the increase in discharge power in the DBD reactors filled with $TiO_2$ and $Al_2O_3$ (Figure 8b). The increase in energy input intensifies the bombardment of the catalyst by the active particles, which results in further desorption of intermediate products on the catalyst [38]. Thus, the number of intermediate products increases in the reaction region, which reduces the number of active substances available for $CO_2$ production and the selectivity of $CO_2$ [40]. In contrast, the $CO_2$ selectivity changed little in the DBD reactor filled with $Fe_2O_3$ (Figure 8b). This is because the $Fe_2O_3$ catalyst has more catalytic activity, and thus has a strong oxidation capacity for naphthalene; this was also confirmed by the mineralization in the DBD reactor filled with $Fe_2O_3$. The discharge wire covered the surface of the electrode and the micro-discharge quantity did not increase significantly when the discharge power increased to a certain value [48]. At this time, the generation of carbon oxides tended to be stable.

### 3.7. The Influence of Relative Humidity on Mineralization and $CO_2$ Selectivity

Figure 9 shows the mineralization and $CO_2$ selectivity as a function of RH at a discharge power of 55 W and an initial naphthalene concentration of 28 ppm. The mineralization slightly increased when RH $\leq$ 5% but decreased with the further increase in RH in the DBD reactor filled with $Fe_2O_3$ (Figure 9). The injection of a small amount of water vapor provides a large amount of •OH for improving naphthalene oxidation [49]. However, undissociated water molecules adhere to the catalyst as RH continues to increase, resulting in a decrease in the number of active sites for naphthalene oxidation and a drop in the formation of carbon oxides [50]. This can be illustrated by the change in $CO_2$ selectivity in the DBD reactor filled with $Fe_2O_3$, as shown in Figure 9b. Although RH had little influence on mineralization, which remained at about 57.9%, the introduction of water vapor promoted the oxidation of CO so that the $CO_2$ selectivity increased in the DBD reactor filled with $Al_2O_3$.

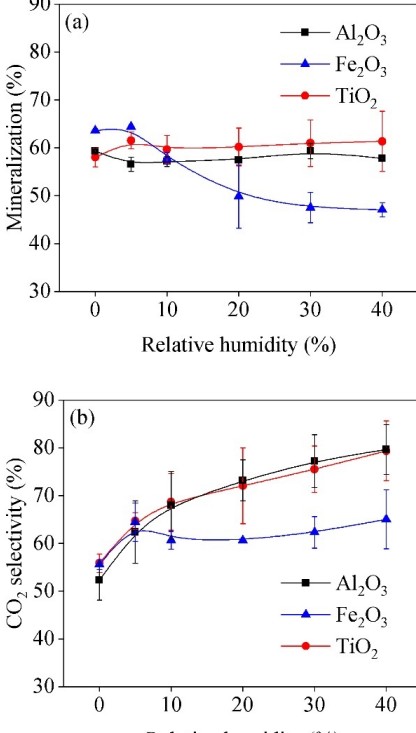

**Figure 9.** Effect of relative humidity on (**a**) the mineralization and (**b**) the $CO_2$ selectivity in the DBD reactor filled with $Al_2O_3$, $Fe_2O_3$, and $TiO_2$. (Discharge power: 55 W; driving frequency: 12.6 kHz; initial naphthalene concentration: 28 ppm; total gas flow rate: 500 mL/min; discharge gap: 3 mm).

It is worth noting that the addition of steam increased the mineralization and the increase was most obvious when RH was 5% in the DBD reactor filled with $TiO_2$. Compared with the DBD reactor filled with $Al_2O_3$, the mineralization of naphthalene in the DBD reactor filled with $TiO_2$ was higher and its $CO_2$ selectivity was also slightly higher. These phenomena can be attributed to the injection of a small amount of water vapor, which provides a large amount of $\bullet OH$ for the reaction. The $\bullet OH$ can be produced via the following reactions [51,52]:

$$H_2O + e^- \rightarrow \bullet H + \bullet OH + e^- \tag{R10}$$

$$O_2 + e^- \rightarrow O^3(P) + O^1(D) + e^- \tag{R11}$$

$$O^1(D) + H_2O \rightarrow 2 \bullet OH \tag{R12}$$

The $\bullet OH$ is highly active in naphthalene decomposition, contributing to enhanced mineralization and $CO_2$ selectivity. Furthermore, although the enhancement of RH causes undissociated water molecules to adhere to the catalyst, $TiO_2$ is a photocatalyst that can effectively use the ultraviolet light generated by the discharge to generate electron–hole pairs [44]. The holes generate $\bullet OH$ with $H_2O$, and the electrons also generate $\bullet OH$ with $O_2$ [53], which promotes the decomposition of water and overcomes the adverse effects of water to a certain extent.

### 3.8. The Characterization of the Catalysts

The three transition metal oxides ($Al_2O_3$, $Fe_2O_3$, and $TiO_2$) were examined by XRD analysis. As presented in Figure 10, diffraction peaks are compared with the Inorganic Crystal Structure Database (ICSD) data to obtain the crystal parameters of the sample surface [54]. The diffraction peaks of the $Al_2O_3$ catalyst can be observed at 2θ of 25.6°, 35.2°, 43.4°, 57.6°, and 76.9°, corresponding to the (012), (104), (113), (202), (116), and (1010) crystal facets. The peaks of the $Fe_2O_3$ catalyst at 33.4°, 35.8°, 49.8°, 54.5°, and 64.4° correspond to (104), (110), (024), (116), and (300) crystal planes. The $Ti_2O$ catalyst shows diffraction angles (25.3°, 33.5°, 48.0°, 62.7°, and 75.0°) which correlate to the (101), (110), (200), (204), and (215) crystal planes [55].

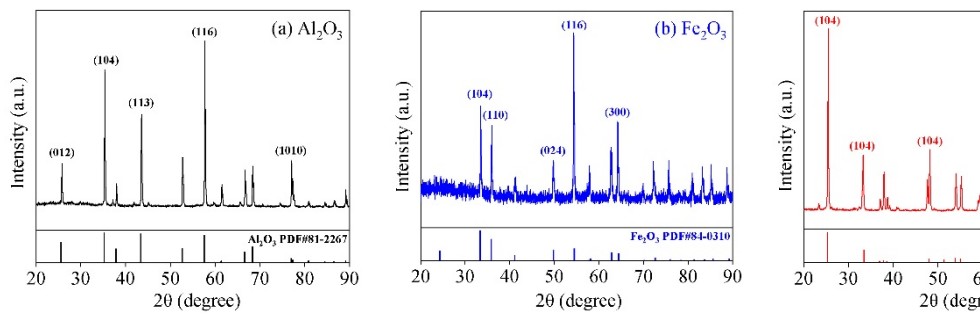

**Figure 10.** X-ray diffraction (XRD) patterns of the (**a**) $Al_2O_3$ catalyst, (**b**) $Fe_2O_3$ catalyst, and (**c**) $TiO_2$ catalyst.

Table 1 shows BET data analysis of the three catalyst samples. The specific surface area, pore volume, and pore size of the catalysts are summarized in the table. The metal oxides having weak adsorption capacity are used for naphthalene degradation, which exclude the influence of naphthalene adsorption on the catalyst. Among these, $Al_2O_3$ and $Fe_2O_3$ have a similar specific surface area and different pore size, and $Al_2O_3$ and $TiO_2$ have a similar pore size and different specific surface area. As can be seen from the table, $Fe_2O_3$ shows a smaller pore size than the other catalysts; thus, the $Fe_2O_3$ catalyst is more negatively affected because undissociated water molecules cover its surface, as shown when exploring the influence of RH (Figure 9a) [56]. In addition, $TiO_2$ shows a higher specific surface area, pore volume, and pore size than the other catalysts; these characteristics are conducive

to the exposure of the active sites, and promote the photocatalytic performance of the catalyst [55].

**Table 1.** BET data analysis of the three catalyst samples.

| Catalysts | Specific Surface Area (m$^2$/g) | Pore Volume (cm$^3$/g) | Pore Size (nm) |
| --- | --- | --- | --- |
| Al$_2$O$_3$ | 0.3056 | 0.000629 | 17.7985 |
| Fe$_2$O$_3$ | 0.7785 | 0.001263 | 7.0580 |
| TiO$_2$ | 6.7880 | 0.025519 | 19.7173 |

BET: Brunauer–Emmett–Teller.

The surface morphology of the three samples was studied using SEM, as shown in Figure 11. The samples have micron-sized pores, as seen from the SEM images. The Al$_2$O$_3$ catalyst possesses a smooth surface and a structure having fewer pores, as shown in Figure 11a,d. The surface of the Fe$_2$O$_3$ catalyst has loose porosity, and its pores are large (Figure 11b,e). The TiO$_2$ catalyst has a coarse surface and a finer porous structure than the other two catalysts (Figure 11c,f). The porous structure of the TiO$_2$ catalyst is conducive to the exposure of the active sites, thus promoting the photocatalytic performance of the catalyst. The analysis of the catalysts' specific surface area also confirms this viewpoint, as shown in Table 1. Thus, TiO$_2$ shows more apparent advantages in terms of the decomposition and mineralization of naphthalene than the other two catalysts.

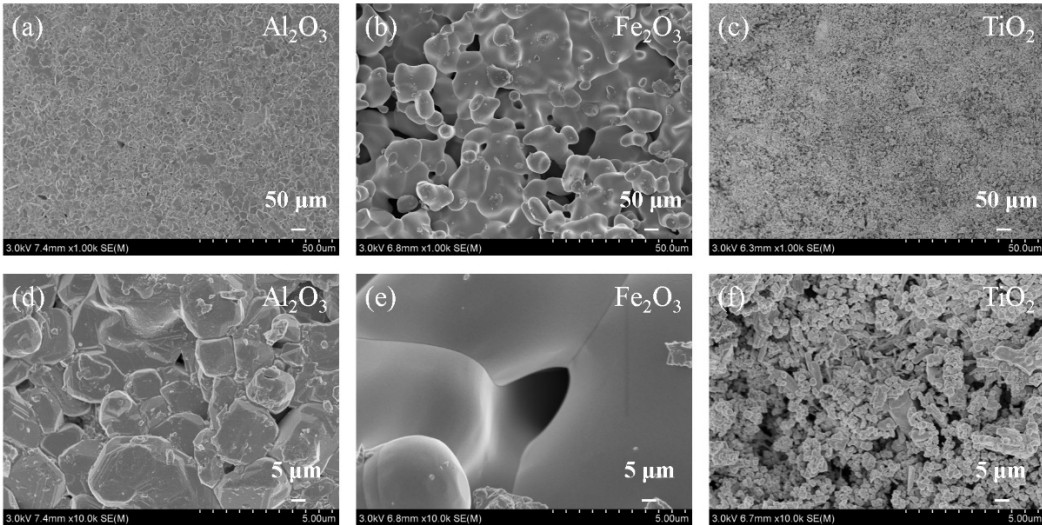

**Figure 11.** The scanning electron microscope (SEM) images of (**a**,**d**) Al$_2$O$_3$ catalyst, (**b**,**e**) Fe$_2$O$_3$ catalyst, and (**c**,**f**) TiO$_2$ catalyst.

## 4. Conclusions

In this study, an AC power supply-excited packed-bed DBD in synthetic air was used to oxidate naphthalene. The results show that the filling materials, initial naphthalene concentration, gas flow rate, discharge power, treatment time, and RH all have certain effects on the decomposition of naphthalene and the generation of carbon oxides. The addition of filling materials reduces the breakdown voltage in the discharge gap and increases the number of micro-discharges. Furthermore, the catalytic performance of the filling materials plays a vital role in naphthalene decomposition. In general, TiO$_2$ has more obvious advantages than the other two catalysts in terms of the reaction because of its photocatalytic properties and large specific surface area. The mineralization reached 82.2% at a discharge power of 55 W when $C_i$ was 21 ppm and the total gas flow rate was 1 L/min in the DBD reactor filled with Al$_2$O$_3$.

As the treatment time increased, the reaction eventually achieved a dynamic equilibrium through the continuous cycle of adsorption and desorption on the catalyst surface

in the discharge process. The discharge wire covered the electrode surface and the micro-discharge quantity did not increase significantly when the discharge power increased to a certain value. The effect of humidity on the discharge was related to the proportion of the introduced water vapor and the type of catalyst in the reactor. $Fe_2O_3$ had a positive effect on the naphthalene oxidation when the amount of water vapor was low, but a negative effect when the water vapor increased, due to its small pore size. The $TiO_2$ catalyst can overcome the adverse effects of water molecule attachment due to its photocatalytic properties and porous structure.

**Author Contributions:** Conceptualization, D.C. and D.Y.; methodology, D.C., Y.L. (Yunhu Liu) and T.F.; software, J.L. and Y.L. (Yueyue Liu); validation, Z.Z. and X.C.; formal analysis, J.L. and Z.Z.; investigation, J.L.; resources, D.C. and D.Y.; data curation, D.C.; writing—original draft preparation, J.L.; writing—review and editing, D.C.; visualization, J.L.; supervision, D.C.; project administration, D.C.; funding acquisition, D.C. and D.Y. All authors have read and agreed to the published version of the manuscript.

**Funding:** This research was funded by the National Natural Science Foundation of China, grant number 51967017; the Science and Technology Plan Project of the Ninth Fund of Xinjiang Production and Construction, grant number 2021JS005, 2021JS003; the Fundamental Research Funds for the Central Universities, grant number RCZK2018C36.

**Data Availability Statement:** Not available.

**Conflicts of Interest:** The authors declare no conflict of interest.

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
