# Peer review of "Decomposition of Naphthalene by Dielectric Barrier Discharge in Conjunction with a Catalyst at Atmospheric Pressure"

_catalysts, doi:10.3390/catal12070740_

Round 1

Reviewer 1 Report

The present manuscript is devoted to use of the (DBD) plasma reactor for decomposition of naphthalene in air cooperating with the metal oxide catalyst.

Nevertheless, the paper is not about catalysis.

“The results show that the filling materials, initial naphthalene 363
concentration, gas flow rate, discharge power, treatment time, and RH all have certain 364
effects on the decomposition of naphthalene and the generation of carbon oxides.”

According to present data, the effect of the filling materials is less remarkable compared with parameters of the process (time, discharge power…)

Only RH is noticeably sensitive to the nature of the filler.

The principle questions:

1.    Why, if TiO2 is the best, all experiments were done only for Al2O3? Since the TiO2 and Al2O3 are different in conductivity, it would be interesting to see the same plots as shown in figure 2 for TiO2. Not enough attention is paid to the difference in conductivity.

2.    If TiO2 photocatalytic properties are important, what is the mechanism? What is the source of the photons/light? Plasma? If so, when TiO2 is used as filler, the regime of plasma generation should be very important. It can be expected that the dependence of the process on the parameters of plasma generation for TiO2 and Al2O3 can be different.

3.    According to the main idea of the work, the phase, porous structure, specific surface area and morphology of the oxide material are very important.

Section 2.1 Dielectric materials. “In this paper, three transition metal oxides (Al2O3, Fe2O3, and TiO2) are used as 89
filling catalysts in the discharge gap. All of these filling materials are commercially 90
available and untreated.” Information is not enough. What was crystal modification of oxides, powder or grains were used. For example, if different commercially available Al2O3 (alpha and gamma) with different size of the grains and specific surface area would be used, can we see the difference? It is important to show that the characteristics of the material have less influence compared with oxides of different nature.

4.    Section 3.8 should be removed. Authors show some data, but did not draw any conclusions. XRD shows the presence of crystal faces. So what (for this work)?

It is not the characterization of the catalysts. 1) XRD does not provide the information about “the sample surface”. XRD provides information about the phase of the material, which is important for this work.

2) “The porous structure of TiO2 catalyst 357
is conducive to the exposure of the active sites, and promotes the photocatalytic 358
performance of the catalyst.”

How the porous structure seen in SEM is connected with photocatalytic activity? Al2O3 can have different porous structures. BET analysis could be better in this case.

5.    “Although there are many studies on the 74
treatment of gas phase pollutants by DBD and catalyst combination technology, there are 75
relatively few experiments on the treatment of gas phase naphthalene by using 76
it. Therefore, it is necessary to study the treatment of naphthalene by combining plasma 77
and catalyst methods.” Are these “relatively few experiments” presented in literature? If so, please, add the literature references. Emphasize, what is new was made in the present work.

I would recommend to the authors to withdraw the manuscript, remove section 3.8., check different oxide materials (grains/powder, crystal modifications) and submit to the engineering journal with optimization of the process.

Reviewer 2 Report

This manuscript is about a very important issue, the removal of polyaromatic pollutants from the atmosphere. The authors do this using Dielectric Barrier Discharges for naphthalene, the simplest and most frequent polyaromatic  in the presence of 3 filling catalytic materials, whose discharge characteristics they have measured. They use these measurements to explain the additional effect of those materials as well as the catalytic effect and the competition between decomposition with further oxidation of decomposition products and oxidation of naphthalene. Furthermore, the authors study the effect of other process variables with a strong discussion of their results.

This experimental work of this novel method of polluntants removal is carefully carried out and the results are reliable and novel. The manuscript is well structured and written with critical discussion of the results.

It should be accepted for publication as it is.

My only point has to do with the title which refers to Decomposition of Polycyclic Aromatic Hydrocarbons, while the authors studied specifically the decomposition of naphthalene. Hence, I think it should be more appropriate if this is reflected in the title: ”Decomposition of Napthalene by …”or ”… Napthalene as a model Polycyclic Aromatic Hydrocarbon …”

After the authors address this, the paper could be accepted for publication.

Reviewer 3 Report

In this manuscript (catalysts-1780529) non-thermal plasma generated by DBD under air atmosphere was applied in combination with metal oxide catalyst for removal PAHs. The topic and objective of this study is suitable for publication in Journal of Catalysts. Overall the quality of English and grammar needs to be thoroughly checked. The length of the manuscript is appropriate. However, the quality of this manuscript should be improved. The authors should address major concerns which I outline below.

1.    The abstract should be presented a summary of the ideas and outcomes described in the paper. The current form needs a major revision. The full names of the abbreviations should be provided when they appear firstly in the text.

2.    In the scientific research paper, we prefer if you use the third person singular, instead of the first person singular or plural (e.g. avoid using 'we', "our" in the manuscript).

3.    Many similar reports have been published on the same topic. What is novelty in this paper? The novelty of this work should be explained in last paragraph of the introduction part.

4.    The value of energy yields is recommended to be calculated and discussed. Please see https://doi.org/10.1016/j.psep.2017.11.005

5.    The introduction section should be expanded. Application of non-thermal plasma for environmental remediation using various reactor design can be discussed and recommended for authors' considerations. Key references:

https://doi.org/10.1016/j.arabjc.2021.103366

https://doi.org/10.1016/j.seppur.2019.01.074

6.    How does the temperature change before and after the discharge? Has the author been tested the temperature?

7.    In DBD plasma the generated ozone is the main reactive species under air atmosphere. Have the authors measured the ozone production during the discharge?

8.    The conclusion is excessively long and may be condensed to make it more informative.

9.    Error bar should be provided because it is important for error analysis of the experimental results.

10. A full proof reading for English needs to be carried out as there are several grammatical and language mistakes though out the text. On some occasions, it was difficult to understand the meaning of some sentences in the text.

Reviewer 4 Report

This work is devoted to the study of the processes of catalytic decomposition of polycyclic aromatic hydrocarbons. For these purposes, dielectrics were used in the work: oxides of iron, aluminum, titanium. The introduction substantiates the topic and the relevance of the work, it is written quite fully. The abundance of experimental data has a good effect on this work. In general, the work is written at a high experimental and theoretical level. The experiment is constructed adequately, and the conclusions are confirmed by experimental data. There are some points for improvement:

1. Figure 1. Is this experimental setup a development of the authors or an already known model? If this is the development of the authors, then it is desirable to indicate this more clearly. This will be a significant plus.

2. When describing the obtained data, it is desirable to use the following publications: 10.1002/cphc.201801160, 10.1002/cphc.202000455. In addition, it is desirable to add more theoretical justifications to the description of the obtained data.

3. Why did the authors choose these dielectrics? In this context, it is desirable to mention the areas of their application as catalysts, in particular, the works: 10.1007/s13399-022-02587-x, 10.1155/2014/727496, 10.1039/C7CC04742H.

4. Conclusions. Very extended. It is desirable to make them more concise.

5. Please add more literature to the part where you describe the experimental data. This will improve the quality of this work.

Round 2

Reviewer 1 Report

As before I think that this manuscript is not about catalysis and should be published elsewhere.

1)

3.8. The Characterization of the Catalysts. 360
Figure 10. X-ray diffraction (XRD) patterns of (a) Al2O3 catalyst (b) Fe2O3 catalyst (c) TiO2 catalyst. 361
Three transition metal oxides (Al2O3, Fe2O3 and TiO2) are examined by XRD analysis. 362
As presented in Figure 10, diffraction peaks are compared with The Inorganic Crystal 363
Structure Database (ICSD) data to obtain the crystal parameters of the sample surface [54]. 364
The diffraction peaks of Al2O3 catalyst can be observed at a 2θ of 25.6â—¦, 35.2â—¦, 43.4â—¦, 57.6â—¦, 365
and 76.9â—¦, corresponding to the (012), (104), (113), (202), (116), and (1010) crystal facets. 366
The peaks of the Fe2O3 catalyst at 33.4â—¦, 35.8â—¦, 49.8â—¦, 54.5â—¦, and 64.4â—¦ correspond to (104), 367
(110), (024), (116), and (300) crystal planes. The Ti2O catalyst shows diffraction angles 368
(25.3â—¦, 33.5â—¦, 48.0â—¦, 62.7â—¦, and 75.0â—¦) which correlate to the (101), (110), (200), (204), and (215) 369
crystal planes [55].

So what? Oxide “shows” some angles, what conclusion can be done for the paper? What crystal modifications you have? XRD is not about surface.

XRD is used for phase identification of crystalline material. Bulk composition can be determined by XRD.

2)

In addition, the TiO2 shows a higher 381
specific surface area, pore volume and pore size than the other two, which is conducive 382
to the exposure of the active sites, and promotes the photocatalytic performance of the 383
catalyst[55].

2.1) Were atypical numbers of Specific Surface Area. Crystal modifications of oxides should be evaluated basing on XRD data.

2.2) So if you will take another Al2O3 with higher SSA (similar to TiO2) you will get another result?

2.3) In addition, the TiO2 shows a higher 381
specific surface area, pore volume and pore size than the other two, which is conducive 382
to the exposure of the active sites, and promotes the photocatalytic performance of the 383
catalyst[55].

SSA of TiO2 is responsible for photocatalytic activity? No, author mentioned:

This may be due to the fact that TiO2 is a photocatalyst, which 282
can use the ultraviolet light generated by the discharge to generate electron-hole pairs that 283
promote the production of various strong active species, such as OH, HO2, and so 284
on[44]. Thus, the decomposition of naphthalene can be promoted. The active species can 285
be produced via the following reactions (see R3-R9) [44, 45].

This is correct. The ultraviolet light generated by the discharge is needed. Why discharge power dependences are similar for all oxides?  

2.4. The 394
analysis of catalyst specific surface area also confirms this viewpoint via the Table 1.

If authors believe that SSA is important for the obtained results, TiO2 samples with different SSA and Al2O3 samples with different SSA should be compared. Such materials are commercially available. This can confirm importance of TiO2.

3. Still very important trends shown on Al2O3. And TiO2 should be added.

Manuscript is not ready for publication in “Catalysts” and should be redirect to elsewhere.

Reviewer 3 Report

 All comments have been addressed properly, so the article is suggested to be published in its present form. A further read throughout to correct minor spelling mistakes is worth doing.